# Anomalous flocking in nonpolar granular Brownian vibrators

Yangrui Chen ![ORCID][1] & Jie Zhang ![ORCID][1,2] ✉

Using Brownian vibrators, we investigated the structures and dynamics of quasi-2d granular materials, with packing fractions ($\phi$) ranging from 0.111 to 0.832. Our observations revealed a remarkable large-scale flocking behavior in hard granular disk systems, encompassing four distinct phases: granular fluid, flocking fluid, poly-crystal, and crystal. Anomalous flocking emerges at $\phi$ = 0.317, coinciding with a peak in local density fluctuations, and ceased at $\phi$ = 0.713 as the system transitioned into a poly-crystal state. The poly-crystal and crystal phases resembled equilibrium hard disks, while the granular and flocking fluids differed significantly from equilibrium systems and previous experiments involving uniformly driven spheres. This disparity suggests that collective motion arises from a competition controlled by volume fraction, involving an active force and an effective attractive interaction resulting from inelastic particle collisions. Remarkably, these findings align with recent theoretical research on the flocking motion of spherical active particles without alignment mechanisms.

Collective motion, characterized by the coordinated behavior of microscopic components on a large scale in both space and time, is a pervasive phenomenon observed in various systems, including soft matter and active matter[1–4]. In the realm of active matter, the mechanisms driving collective motion have been extensively studied. For instance, the collective behavior of bird flocks can be effectively modeled as a system of "flying spins", where the alignment of neighboring individuals' moving directions plays a crucial role[5–7]. Similarly, rod-like particles can be described as active nematics, incorporating factors such as local alignment and volume exclusion to explain their collective motion[8]. Notably, when particles possess intrinsic spinning motion, intriguing topologically protected edge modes can arise due to nonreciprocal interactions[9]. These examples highlight the profound understanding gained in elucidating the mechanisms governing collective motion in active matter systems.

Collective motion in granular materials is often associated with phenomena such as jamming, particle polarity, and particle shape. In densely packed granular systems subjected to quasi-static shear, the behavior of contacts and contact forces plays a crucial role in floppy modes[10,11], plastic deformation[12,13], and the formation of turbulent-like vortices[14,15]. When granular materials are subjected to vibration, inelastic collisions become prominent, and the polarity and shape of the particles become significant factors. Systems composed of self-propelling polar particles[16–18] and self-spinning disks[19,20] exhibit collective behaviors, blurring the line between granular materials and active matter. In addition, the behavior of rod-like particles can be effectively described using the framework of active nematics[8,18,21–23]. Simulations have shown that the alignment interaction plays a critical role in the flocking behavior of self-propelling purely repulsive hard disks[24]. In the absence of this alignment interaction, only motility-induced phase separation is observed[25,26]. Previous experiments with purely repulsive hard spheres or disks, lacking preferred translational or rotational directions, under uniform driving in a 2d system, have not exhibited any flocking motion[27–31].

However, recent theoretical work by Caprini et al.[32] challenges the prevailing notion that alignment interactions are a prerequisite for the emergence of flocking behavior. Their study predicts that flocking can occur even in the absence of explicit alignment interactions. Instead, the transition between disordered and flocking phases is primarily determined by the interplay between the active forces and the attractive forces arising from neighboring particles. Since inelastic collision is analogous to an effective attraction, conducting

¹School of Physics and Astronomy, Shanghai Jiao Tong University, Shanghai, China. ²Institute of Natural Sciences, Shanghai Jiao Tong University, Shanghai, China. ✉e-mail: jiezhang2012@sjtu.edu.cn

[1]: #

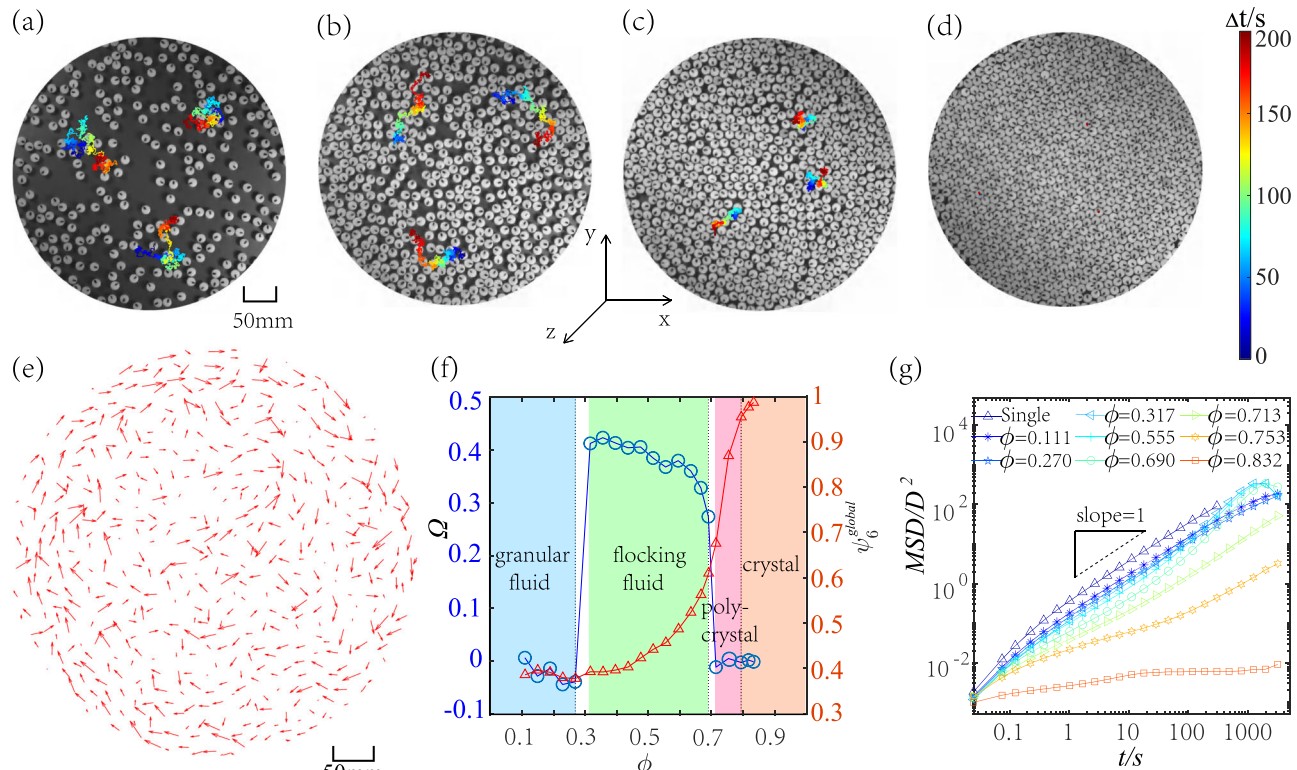

**Fig. 1 | The dynamics of non-polar granular Brownian vibrators. a–d** Snapshots of particle configurations at $\phi$ = 0.270, 0.555, 0.713, 0.832, respectively. The trajectories of three particles are depicted on each snapshot with color bars ranging from blue to red representing 0–200 s. **e** Particle displacement field at $\phi$ = 0.555. The length of the arrows is scaled to half of the displacement to enhance visibility. **f** Granular fluid ($\phi \leq 0.270$), flocking fluid ($0.317 \leq \phi \leq 0.690$), poly-crystal ($0.713 \leq \phi \leq 0.753$), and crystal ($\phi \geq 0.793$). $\Omega$(blue circle) is the average curl of the particle displacement field within $\Delta t$ = 100 s. $\psi_6^{global}$ (red triangle) is the spatio-temporal average of the absolute value of hexatic order parameter $\psi_6$. **g** MSD of the particle's translational motion. At $\phi$ = 0.317, 0.555, 0.690, the MSD peaks around 2000 s correspond to a half period of the collective rotation of the outermost particles. These particles contribute the most to the MSD and travel from one side of the system to the other through an approximately semicircular arc path, reaching the system size scale. Source data are provided as a Source data file.

experiments with vibrated granular materials may provide a promising avenue for experimentally verifying this prediction, which remains unverified to date.

In this paper, we present a systematic investigation of the structures and flocking dynamics in quasi-2d experiments using Brownian vibrators. These vibrators consist of disks with alternating inclined legs along the rim. By exploring a wide range of packing fractions ($\phi$), we uncover a rich variety of structures and dynamics arising from inelastic particle collisions. Specifically, we observe the emergence of distinct phases, namely the granular fluid, flocking fluid, poly-crystal, and crystal. Interestingly, while the poly-crystal and crystal phases exhibit striking similarities to equilibrium hard disks[33–38], the first two phases differ significantly from both the equilibrium systems and previous quasi-2d experiments involving uniformly driven spheres[27,28,39–51].

Remarkably, our experiments demonstrate that particles exhibiting stochastic motion on short time scales manifest flocking around the center of the system on longer time scales within the range of $0.317 \leq \phi < 0.713$. The critical packing fraction of 0.713 is close to the crystallization point observed in the early experiment[28], while the critical packing fraction of 0.317 aligns with the flocking transition point calculated by the simplified dynamical equations for non-polar active particles[32]. We experimentally verified that active particles can exhibit flocking behavior even in the absence of polar alignment interactions. Overall, our investigation reveals that granular materials subjected to uniform stochastic driving exhibit weak cohesion, intricate internal structures, and dynamic behaviors. Furthermore, our findings highlight the possibility of large-scale collective motion in purely repulsive hard-disk systems, broadening our understanding of the collective dynamics in granular materials.

## Results

In the pre-experiment, each individual particle exhibits Brownian-like behavior, characterized by uncoupled translation and rotation with Gaussian distributions centered around zero means, in the absence of particle-particle collisions[52].

Figure 1 provides snapshots of particle configurations representing the (a) granular fluid, (b) flocking fluid, (c) poly-crystal, and (d) crystal phases. The trajectories of three particles are plotted to illustrate their different dynamics. At $\phi$ = 0.270, driven by the random forces generated from collisions between the bottom plane and tilted legs of the particles, the particles perform random walks at all times in Fig. 1a.

At $\phi$ = 0.555, the particles traverse a clockwise arc around the center of the plane in Fig. 1b. The displacement field over a time interval of 100 s is depicted in Fig. 1e as a giant clockwise vortex, suggesting a unidirectional motion on long timescales. Notably, on short time scales ($\Delta t < 10$ s) in Fig. 1b, individual particles are still dominated by random driving from the bottom plane, and the paths remain approximately random walks, which qualitatively differs from the phenomena of particle orientation and instantaneous velocity alignment reported in previous experiments[16–18]. In the "Discussion" section, we will elaborate in detail on the reasons why the flocking fluid phase has not been observed in previous literature. The large-scale collective motion observed at $0.317 \leq \phi \leq 0.690$ can be characterized by the nonzero vorticity $\Omega$—the average curl of the particle displacement field (Fig. 1f and Supplementary Methods 1.5 in Supplemental Information (SI)). Although all observed flocking directions are clockwise, the disappearance of flocking at $\phi \leq 0.270$ implies that the density-dependent, nontrivial phenomenon is not solely generated by

the setup of fixed boundaries and inherent asymmetries in our experimental system.

The flocking behavior disappears at $\phi \geq 0.713$ with $\Omega$ rapidly drops to 0 in Fig. 1f. At $\phi = 0.713$, particles diffuse around slowly in Fig. 1c. At $\phi = 0.832$, the particles oscillate near the lattice like a crystal in Fig. 1d. The global mean hexagonal order parameter $\psi_6^{global}$ (Fig. 1f) and Supplementary Methods 1.3 in Supplemental Information) marks the phase transition between poly-crystal ($0.67 \leq \psi_6^{global} \leq 0.86$) and crystal phases ($\psi_6^{global} > 0.95$). The crystallization above $\phi = 0.713$ in Fig. 1f is similar to the early experiment[28], where spheres were sandwiched and vertically vibrated between two horizontal plates, and their system shows subdiffusive, caging-type behaviors on MSD at intermediate time scales at $0.652 < \phi < 0.719$, similar to the curve of $\phi = 0.690$ in Fig. 1g. However, within $0.652 < \phi < 0.719$, no large-scale motions are observed[28]; a so-called "isotropic fluid phase" was observed for $\phi < 0.652$[28,53]. Moreover, our system is locally more ordered as shown in Supplementary Figs. 5 and 6. Interestingly, $\phi = 0.713$ of poly-crystal in Fig. 1f is nearly identical to the melting-transition point $\phi_s = 0.716$ predicted for the equilibrium hard disks[33]. Furthermore, $\phi = 0.690$ of flocking fluid in Fig. 1f is identical to the value of the pure fluid $\phi_f$[33]. Note that the precise values of $\phi_s$ and $\phi_f$ may vary slightly[33–38]. Although there is no data point within $0.690 < \phi < 0.713$ due to the discrete increment of $\phi$, the particle configurations at $\phi = 0.690$ and $0.713$ show different symmetries in Supplementary Fig. 7.

The translational mean square displacement (MSD) reveals the dynamic characteristics of different phases in Fig. 1g. For $\phi \leq 0.270$, the particles exhibit superdiffusive behavior for $t \lesssim 0.2$ s before transitioning to normal diffusion, indicating that a particle persistence time of $\tau_p = 0.2 \pm 0.05$ s at $\phi \leq 0.270$, as shown in Supplementary Fig. 2. For $\phi = 0.317, 0.555, 0.690$ in the flocking fluid phase, particles are superdiffusive for $t \lesssim 0.1$ s before turning subdiffusive for $0.1$ s $\lesssim t \lesssim 20$ s. Due to the flocking on long timescales, superdiffusion appears again for $t \gtrsim 20$ s. In the poly-crystal phase, the MSD of $\phi = 0.713$ is divided into three parts: the superdiffusion for $t \lesssim 0.05$ s, the subdiffusion for $0.05 \lesssim t \lesssim 30$ s, and the diffusion for $t \gtrsim 30$ s due to particles at grain boundaries (see Supplementary Figs. 4 and 5). As $\phi$ increases to $\phi = 0.753$, the MSD slope gradually decreases and eventually down to 0 till $\phi \geq 0.793$ in the crystal phase. The self intermediate scattering functions $F_s(q, t)$ (see Supplementary Fig. 11) also show the diffusive properties of particles at different time scales, which are completely consistent with the results from MSD.

Figure 2 plots the pair correlation functions $g(r)$ at $\phi = 0.111$ (For more $g(r)$ at different phases, please refer to Supplementary Discussion 2.4 in SI), whose first peak corresponds to the maximum possible center-to-center distance $d_1$ between neighboring particles. In Fig. 2, the peak of $g(r)$ is notably higher than that of equilibrium hard disks[54]. This disparity arises due to the dissipative nature of inelastic collisions, which allows particles to approach each other more closely than they would in equilibrium. For a detailed comparison of $g(r = D)$ versus $\phi$ with equilibrium hard disks and early experiments[53], please refer to Supplementary Fig. 1.

Assuming that the inelastic collisions can be effectively represented by an attractive potential[55–58], according to Boltzmann's law, the $g(r) = \exp(-V(r)/(kT)) \cdot g_{eq}(r)$. The equilibrium $g_{eq}(r)$ for the purely repulsive hard spheres is obtained from the BGY theory in ref. 54. The interparticle force $F_i = -\frac{d(V(r))}{dr}$ and its maximum:

$$F_{max} = \frac{1}{2} m v_T^2 \left[ \frac{d \ln\left(\frac{g(r)}{g_{eq}(r)}\right)}{dr} \right]_{max}. \qquad (1)$$

Here the thermal velocity $v_T = 1.4 \pm 0.1 D \cdot s^{-1}$ is weakly affected by $\phi$[52]. Inspired by the theoretical work by Caprini et al.[32], the simplified

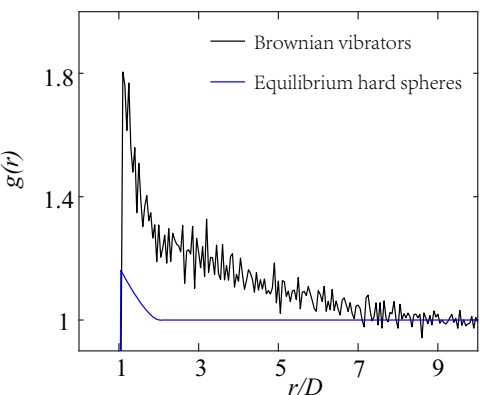

**Fig. 2 | Anomalous first peak of the pair correlation function.** The pair correlation functions of extremely dilute granular fluid at $\phi = 0.111$. The black line is the experimental results of our Brownian vibrators. The blue line is the numerical solutions of the BGY theory in equilibrium hard disks[54]. Source data are provided as a Source data file.

dynamical equations for non-polar active particles can read:

$$m\dot{\mathbf{v}}_\mathbf{i} = -\gamma \mathbf{v}_\mathbf{i} + \mathbf{F}_\mathbf{i} + \gamma_0 v_0 \mathbf{n}_\mathbf{i} + \sqrt{2\Lambda\gamma}\boldsymbol{\eta}_\mathbf{i}, \qquad (2)$$

$$\dot{\boldsymbol{\theta}}_\mathbf{i} = \sqrt{2D_r}\Theta_i. \qquad (3)$$

Here $\boldsymbol{\eta}_\mathbf{i}$ and $\Theta_i$ are white noises with zero average and unit variance, $\Lambda$ is a constant, and $m$ refers to the particle mass. The damping coefficient $\gamma$ describes the dissipation of kinetic energy. $\mathbf{F}_\mathbf{i}$ represents the net force on the particle $i$ from all other particles in the system. In our model, it is averaged to the effective attraction corresponding to the energy loss due to inelastic collisions between particles.

Unlike the simulation where the active force is given as $\gamma v_0$[32], in our experiment, the magnitude of the active force $\gamma_0 v_0$ originates from the collisions between the bottom surface and each inclined leg with a unique structure (see "Methods" section for details).

It's worth mentioning that in the polar case, the direction of the active force $n_i$ coincides with the particle's orientation, which is identified by the asymmetric shape of the disk's legs[16]. In contrast, in the non-polar case discussed in our experiments, the direction $\mathbf{n}_\mathbf{i}$ of the active force is related to the specific impact kinematics between the randomly oriented legs of the disk and the bottom plate, and determined by an inclined leg which is instantaneously propelled after contacting the bottom surface. Due to the randomness of the contact angle between the inclined legs and the bottom surface, the direction $\mathbf{n}_\mathbf{i}$ of the active force is independent of the velocity direction $\frac{\mathbf{v}_\mathbf{i}}{|\mathbf{v}_\mathbf{i}|} \equiv (\cos \theta_i, \sin \theta_i)$. $\mathbf{n}_\mathbf{i}$ changes at a high frequency with the collisions between the particles and the bottom surface. Thus, in the non-polar case, a single particle can experience an active force with different orientations without rotating if the impact with the plate involves differently oriented legs.

And the diffusion coefficient $D_r$ is approximated from the diffusion coefficient of the particle velocity direction $\frac{\mathbf{v}_\mathbf{i}}{|\mathbf{v}_\mathbf{i}|}$ in experiments. The active velocity $v_0$, and the friction factor $\gamma_0 \equiv m/\tau_I$ can all be determined experimentally in Supplementary Methods 1.1 in Supplemental Information, where the $\tau_I$ is the inertia time. The Péclet number $Pe = v_0/(D_r D)$ is viewed as the ratio between persistence length and particle size, quantifying the distance over which a particle maintains directional persistence.

Comparing the maximum equivalent attractive force and the active force, the critical value $Pe_c = F_{max}/(\gamma_0 v_0) \cdot Pe$ is positively correlated with the collision frequency through $F_{max}$. As shown in Fig. 3, $Pe$ decreases with $\phi$ since the persistence time $\tau_p = 1/D_r$ decreases with $\phi$.

Meanwhile, the collision frequency increases with $\phi$, so the maximum equivalent attractive force $F_{max}$ increases with $\phi$. At $\phi \le 0.270$, $Pe$ is greater than $Pe_c$, indicating that the active forces dominate over the attractive forces, leading to the random motion. At $0.317 \le \phi \le 0.690$, where the maximum attractive force exceeds the active forces, long-time scale collective motion emerges. For specific details, please refer to Supplementary Methods 1.1 in Supplemental Information.

Compared to the theoretical work by Caprini et al.[32], where collective behavior emerges when the maximum force exerted by neighboring particles surpasses the active forces of the individual particles, leading to global polarity in the instantaneous velocity directions, our experiments did not observe the alignment effect of the instantaneous velocity directions of particles, indicating a significant distinction between the global flocking we observed and the collective behavior of elongated particles or particles subject to alignment interactions observed in previous studies[8,24]. In the simulation[32], the flocking phase exists under overdamped conditions, whereas our experiment is conducted under underdamped conditions, making our work somewhat different from their simulation results. In addition, due to the periodic boundary conditions in the simulation, the collective behavior appears as transitional flocking. In our quasi-2d system with fixed boundaries, the collective behavior manifests as global collective rotation around the center of the system. However, the flocking transition point calculated using the simplified dynamical equations for non-polar active particles aligns with the critical packing fraction observed in our experiments. Our experimental results confirm the existence of flocking behavior in non-polar active matter in the absence of alignment interactions as predicted by Caprini et al.[32].

The local packing fraction of disks with diameter $D$ is determined by the formula $\phi_{local} = \frac{\pi D^2}{4 S_v}$, where $S_v$ denotes the area of the corresponding Voronoi polygons as shown in Supplementary Fig. 4. The distribution of $\phi_{local}$ does not exhibit clear bimodal peaks at any given $\phi$. This observation suggests that in a spatially homogeneous driven quasi-2d granular system, the typical clustering or solid-liquid phase separation observed in previous boundary-driven systems is not present[59]. These findings are consistent with previous experiments involving spatially homogeneous driving conditions[60]. Here the energy is constantly injected and dissipated, resulting in a driven steady state where neighboring particles do not stay in close contact due to stochastic driving separating them, in contrast to free cooling systems[61,62]. Likewise, in the phase-separated systems[27,47–51,63], segregated cluster particles are in close contact, forming a highly compact solid due to the velocity-dependent energy injection rate. Figure 2 provides microscopic physical evidence that dry granular materials subjected to stochastic driving form weak-cohesion fluids at sufficiently low densities, consistent with the early experiments of the spinodal phase separation[50] and the capillary-like interface fluctuations[51] in cohesionless granular systems.

A noteworthy finding is the consistent increase in the fluctuations of $\phi_{local}$ with $\phi$ in the granular fluid phase, as illustrated in Fig. 4b. This increase culminates in a peak at the onset of collective behavior, precisely at $\phi = 0.317$. Subsequently, in the flocking fluid phase, a monotonic decrease in the fluctuations of $\phi_{local}$ is observed. This behavior aligns precisely with the phase transition point corresponding to the emergence of long timescale collective behavior observed in the dynamics of the system.

## Discussion

It's worth mentioning that the flocking observed in our system is different from the collective motion observed in previous experiments with active particles. Over short time scales, the collisions between Brownian vibrators are rare, and the effective mutual attraction is extremely weak, with the active forces dominating, resulting in the particles moving randomly. Only over longer time scales, as the total number of collisions between particles increases and the effective mutual attraction gradually exceeds the active forces, the system exhibits flocking behavior. In our analysis, the effective mutual attraction potential, calculated using the pair correlation function at steady state, results from the accumulation of numerous particle

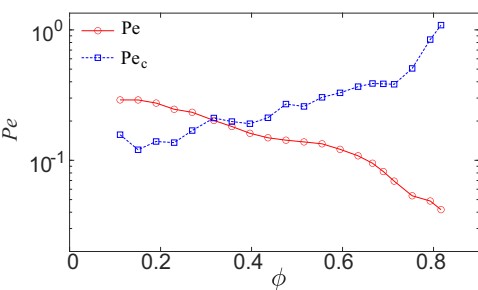

**Fig. 3 | The competition between active forces and effective attractive forces is characterized by two Péclet numbers.** The Péclet number $Pe$ of active particles and the critical value $Pe_c$, which intersect around the phase transition point between granular fluid and flocking fluid phases. Source data are provided as a Source data file.

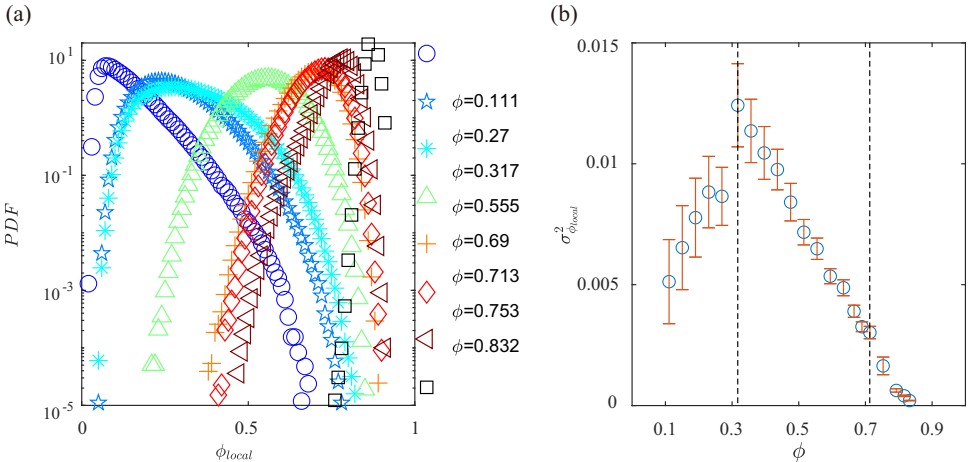

**Fig. 4 | The statistics of local packing fraction. a** The probability distribution function (PDF) of the local packing fraction. **b** The fluctuation of $\phi_{local}$. The error bars represent the standard deviation of 10 sets of data. Source data are provided as a Source data file.

collisions over a long period. Therefore, the competition between the calculated effective mutual attraction force and the active force determines the emergence of collective motion over long-time scales, and is unrelated to the behavior of particles on short time scales.

The granular and flocking fluids phases were not observed in previous works[27,28,39–51], where a single or a shallow layer of spheres is confined within a quasi-2d cell and subject to vertical mechanical vibration. In refs. [41–43], electrostatic and magnetic dipole forces introduce long-range interactions that differ substantially from our experiments. The experiments[44–46] focus on binary mixtures, different from our identical particles. In refs. [28,39,40,47–51], there are only short-range repulsion and inelastic collisions between grains. There are some subtleties between our system and the previous experiments. The main issue is the randomization of particle motion at the single-particle level: using a flat bottom plate[27,47–51] or a cover[64] introduces a non-Gaussian velocity distribution of a single particle, implying spatial correlations of particle movement, which cannot be eliminated with a rough plate or lid[28,39]. The lack of Gaussian statistics in the single particle could induce phase separations[27,47–51] due to a velocity-dependent energy injection rate[65,66]. The phase separation is absent when subject to random forcing in the simulation[65], consistent with our observations. Before our experiments, two attempts were made to ensure Gaussian statistics of velocities[60,67]. However, additional effects were introduced, such as the motions of dimers and the continuous particle rotations along a single direction. Therefore, the Gaussian velocity in the single particle driving is crucial.

In the 3d vertically vibrated hard sphere experiment[68] and accompanying simulations[69], long-term superdiffusion of the detector ratchet angle was observed at high concentrations, which was thought to be linked to the asymmetric defects of the structure as suggested by a simplified simulation model to be associated with the direction of collective motion[70]. However, in our experiments, spontaneous collective behavior emerged due to the competition between active forces due to stochastic driving and the equivalent attraction due to inelastic collisions. Nevertheless, the direction of flocking may be related to the inherent asymmetry in the experimental design, although it is not the root cause of the collective motion, as particles would exhibit collective motion at all $\phi's$ otherwise.

Using Brownian vibrators, our study reveals the existence of four distinct phases in granular materials: granular fluid, flocking fluid, polycrystal, and crystal. Notably, the granular fluid phase showcases the intriguing behavior of purely-repulsive hard disks, where weak cohesion arises from inelastic collisions, resulting in liquid-like structures with fascinating dynamics. A remarkable large-scale collective motion emerges at $\phi = 0.317$, coinciding with the peak in the fluctuations of local packing fractions. This collective motion persists until it terminates near $\phi = 0.713$ where the system crystallizes. Our investigation demonstrates that granular materials subjected to uniform stochastic driving exhibit weak cohesion, complex internal structures, and, notably, a purely repulsive hard-disk system is capable of producing large-scale collective motion.

## Methods

The schematic of the quasi-2D vibration experimental setup is shown in Fig. 5. The vibration driving system, indicated in black, consists of a vertically driven electromagnetic vibrator, an aluminum-magnesium alloy pedestal, an aluminum alloy plate, a flower-shape acrylic boundaries, and a horizontal layer of quasi-2d monodisperse Brownian vibrators. The horizontal adjustment system, marked in red, comprises four vibration isolation airbags located at the bottom of the setup. The optical imaging system, represented in blue, includes a CCD camera and related aluminum profile frame.

The electromagnetic vibrator, fixed to the aluminum-magnesium alloy pedestal, constitutes a vertical vibration platform that provides sinusoidal vibrations along the vertical direction. The aluminum-

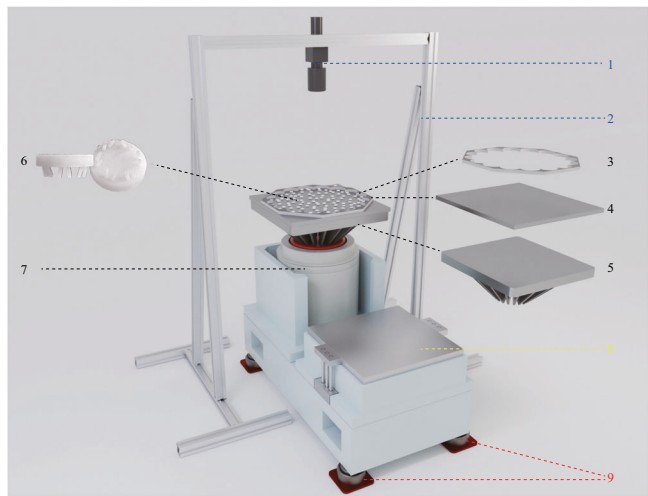

**Fig. 5 | Schematic of experimental setup.** 1. CCD camera; 2. Aluminum profile frame; 3. Flower-shape acrylic boundaries; 4. Aluminum alloy plate; 5. Aluminum-magnesium alloy pedestal; 6. Brownian vibrators; 7. Electromagnetic vibrator; 8. Horizontal sliding stage; 9. Vibration isolation airbag.

magnesium alloy pedestal weighs 60 kg and has a truncated pyramid shape, with its upper surface being a 60 cm × 60 cm plane. The unique microstructure of this aluminum-magnesium alloy provides appropriate vibration damping, reduces the amplification factor of high-frequency resonances, effectively suppresses potential surface standing waves, and thereby enhances the spatial uniformity of the vibration drive. A flat aluminum alloy plate (60 cm × 60 cm × 1 cm) is mounted onto the pedestal by eight strong F-clamps. To prevent particle motion along the boundary, we utilize a flower-shaped acrylic confining boundary[52]. Boundary particles consisting of three layers are excluded from the analysis.

The Brownian vibrator, with a diameter of 16 mm, is 3d-printed using CBY-01 resin with the density $\rho = 1.14 \, \text{g} \cdot \text{mm}^{-3}$, Poisson's ratio $\nu = 0.39$ and Young's modulus $Y = 2$ GPa. Each Brownian vibrator features 12 legs. The legs are 3 mm high and bent inward at an angle of 18.4°, alternately deviating from the mid-axis plane by ±38.5°, as depicted in Fig. 5d. Particle collisions are inelastic, with a restitution coefficient of approximately $\epsilon \approx 0.39$ (For measurement details, please see Supplementary Methods 1.1 in Supplemental Information). The static friction coefficient between the particles and the base is -0.42. Due to the significantly longer time intervals between interparticle collisions compared to the persistent time (detailed in Supplementary Fig. 12), the stochastic driving of the bottom surface on a given particle is minimally affected by the presence of neighboring particles.

The isolation airbags not only reduce the impact of mechanical vibrations on the optical imaging system but also serve as a means to adjust the levelness of the experimental platform. All data analyzed in this study were collected from experiments with continuous vibrations lasting over 20 h, during which particles exhibited no significant gravity-driven drift. This ensures the isotropy of the driving force experienced by the particles. Precise adjustment and careful examination guarantee the stability of the experimental platform under vibration conditions, providing a solid foundation for data analysis.

To obtain the initial state at a specific packing fraction $\phi$, we randomly place particles and allow the system to run for 2 h. The vibration is applied at a frequency of $f = 100$ Hz with a maximum acceleration of $a = 29.4 \, \text{m} \, \text{s}^{-2}$. The resulting amplitude, given by $A \equiv a/(2\pi f)^2 = 0.074$ mm, ensures a negligible vertical displacement of the particles. We capture particle configurations using a CCD camera at a frame rate of 40 frames per second for an hour, enabling further analysis and processing of the data.

The active force on our non-polar active particles is generated by collisions between the particle's tilted legs and the vibrating bottom plane. The unique design of the legs and the presence of noise cause the particle's central axis to slightly tilt away from the gravitational direction during vibration. This results in only a portion of the legs (commonly observed in experiments as a single leg) being instantaneously propelled upon making contact with the bottom surface. Such interactions generate a driving force that accelerates the particle in the horizontal direction, which is influenced by both the velocity at which the collision occurs and the angle of contact between the leg and the bottom surface. The tilted legs leads to a minimal time correlation of the contact angle, resulting in a random orientation of the driving force with a magnitude of $\gamma_0 v_0$ (Supplementary Methods 1.1 in Supplemental Information). The randomness of the driving force direction is precisely the reason why both the translational and rotational velocities of a single Brownian vibrator are Gaussian distributed with zero mean. The key distinction between this active driving force and Brownian noise lies in the orientation of one leg of the non-polar particle during each collision with the bottom plane, with the collision interval ~38 ms and the direction switching frequency of the driving force close to our time resolution of 25 ms. Therefore, the particle is driven by a fixed-magnitude, high-frequency and randomly orientation-switching active force, rather than the environmental Gaussian noise of passive particles.

## Data availability
The datasets generated and/or analyzed during the current study are provided in the Source data file and Supplementary Information. The relevant data on particle positions and velocities can be accessed through the following link in the public database Figshare: https://doi.org/10.6084/m9.figshare.26172586.v1[71]. Source data are provided with this paper.

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

## Acknowledgements
Y.C. and J.Z. acknowledge the NSFC (No. 11974238 and No. 12274291) support and the Innovation Program of Shanghai Municipal Education Commission under No. 2021-01-07-00-02-E00138. Y.C. and J.Z. also acknowledge the support from the Shanghai Jiao Tong University Student Innovation Center.

## Author contributions
Y.C. performed the experimental studies, carried out the analysis and wrote the manuscript. J.Z. supervised this work and wrote the manuscript.

## Competing interests
The authors declare no competing interests.
