## [Peer Review File · Nature Communications]

REVIEWER COMMENTS

Reviewer #1 (Remarks to the Author):

In this paper, the authors are studying a system of granular particles under vertical vibrations. Different regimes of packing fraction are explored and reveal the occurrence of four distinct phases: granular fluid, collective fluid, poly-crystal, and crystal. Poly-crystal and crystal phases do not differ from equilibrium. In contrast, granular fluid and collective fluid phases are characterized by collective motion and, thus, have no equilibrium counterpart. The authors explain collective motion by noting that inelastic collisions can be roughly approximated by attractive interactions and an active force. In this way, the authors claim to have experimentally verified the numerical results of previous papers.

In my opinion, the findings of the paper are interesting: the experimental realization of collective motion numerically observed in a previous paper constitutes a novel result, while polycrystalline and crystalline phases show findings that are rather expected. However, the sections presenting the results are technical and not easy to read. They contain several details that do not easily allow the reader to focus on the novel findings of the paper. In addition, the authors should further investigate the link between their system and the active model employed in previous studies.

A consistent major revision is needed. To be suitable for publication in Nature Communications, the paper should be rewritten to increase its clarity mainly to focus on the collective motion observed at intermediate packing fraction.

I list below several points that the authors should discuss in the new version of the paper.

- in Sec. 2.1, the authors discuss the dynamics of the system for different values of the packing fraction. I have found this section unclear. Indeed, the authors start with a long description of the super-diffusive and subdiffusive behavior by varying the packing that I have, honestly, found super detail. Then, the authors continue the paragraph by quickly introducing the vorticity of the system that is used to show the presence of collective motion. While this measurement is fundamental to distinguishing between collective motion and arrested states, the authors do not focus on this point enough. By summarizing, the authors should carefully describe Fig.1(f) which contains the central result of the paper, i.e.the distinction between the four states and, in particular, the presence of the collective fluid. The authors should describe how the distinction between the four phases is obtained. I suggest a complete rewriting of Sec. 2.1.

While Omega is used to distinguish between the granular fluid phase and collective fluid, the criterion to distinguish between polycrystalline and crystalline phases is not clearly discussed.

- In the literature, there are several studies based on standard granular particles, i.e. non-active. In the introduction, the authors claim that their findings are fundamentally different from those obtained in previous studies. The authors should explain the reason behind this discrepancy.

- In Sec.2, the authors wrote that a single-granular particle displays passive Brownian motion characterized "by uncorrelated translation and rotation with Gaussian distributions centered around zero means" as in their previous study. How does this previous result match with the notion of persistence time that is introduced in Section 2.1?

- The authors are proposing that inelastic collisions can be mapped onto the combination of an effective attraction and a persistent active force. Could the author further comment and intuitively justify this approximation? This is an important point that deserves to be addressed numerically. The authors should discuss this approximation more in detail, by citing references (if possible) and by showing additional studies to compare the dynamics of attractive active particles and granular passive particles with inelastic collisions. This point could be addressed numerically and it is fundamentally to claim the occurrence of the collective motion obtained in previous studies with active particles.

- The authors of Ref.[42] termed the collective motion observed as "flocking". Since the authors are claiming to have experimentally reproduced such a result, I suggest they use the same nomenclature. Note that, in the literature, there are other examples of experimental active granular particles where flocking is claimed even if the system is confined in a box.

- The authors comment on the absence of phase separation in their system. Is this interpretation consistent with the model assumed to understand collective motion? In other words, is the effective Péclet number smaller than the critical Péclet number showing phase separation in systems of underdamped active particles?

- The last paragraph of Sec. II describes the results of the intermediate scattering function at different packing fractions. This paragraph contains several technical details that should be moved in the methods. In addition, the conclusions obtained by this analysis are not clearly explained.

Minor points:

- Intro: Letter  paper.
- Experimental points in Fig.1 (f) are barely visible. Please, modify the figure to fix this problem.

Reviewer #3 (Remarks to the Author):

The submitted manuscript reports an experimental work in which the behavior of a vertically vibrated quasi-2d granular system is explored for a wide range of packing fractions ϕ . Four different phases are observed going from low to high ϕ . The authors refer to them as granular fluid, collective fluid, poly-crystal and crystal. The paper provides an extensive characterization of the dynamics and the structure of all these phases. Among them, the main focus is given to the collective fluid where a global collective rotation is observed and put in relation with recent theoretical/numerical results on active Brownian particles. An important claim of the submitted manuscript is that this collective motion can be explained by the interplay between effective attraction and active forces. While I found the experimental methodology of this work well grounded and correctly explained, I don't find that it provides enough evidence for the observation of collective motion strictly due to the interplay between effective attraction and active forces. Thus, I can't recommend this manuscript as a publication in Nature Communications. My major concern is related to the link the authors make with the theoretical work of Caprini and Löwen [1].

1 Major comments

1.1 Connection with active systems

The authors often refer to their system as an experimental realization of the numerical/theoretical work of Caprini and Löwen [1]. In this reference, it is shown that attractive interactions and active forces are minimal ingredients for the emergence of collective motion (i.e. flocking) characterized by long-range

spatial correlations of velocity polarization. One important message of this article is that explicit alignment interactions are not needed to observe flocking.

In the submitted manuscript, the authors explain that the granular system considered in their experiment, which is composed by Brownian vibrators (i.e. inelastic disks in contact with the vibrating plate through 12 randomly tilted legs), share the same key-properties of the active Brownian particles interacting

via attractive potential used in [1]. An effective attraction between grains results from inelastic collisions, a persistent active force is generated through the friction between the particle and the platform due to the unique structure of the inclined legs of the Brownian vibrators.

I agree that both these key-properties can potentially emerge from the mechanisms highlighted by the authors however I have two objections:

a) It is generally accepted that inelastic collisions are equivalent to an effective attraction but they also introduce in the system an effective velocity

alignment mechanism. Indeed, two colliding inelastic grains mainly lose energy along the direction of the unit vector that connects the particle centers.

Thus, after a collision, the velocities of the two grains result more aligned than

before. This mechanism can be the origin of spatial velocity correlations in

granular systems [2, 3, 4, 5]. The intrinsic presence of effective velocity alignment interactions in granular dynamics is in contradiction with the claim of the

authors that the emergence of flocking in their system is only due to effective attraction and active forces as in Ref. [1]. I think that this is a crucial point that

should be discussed in the main text of the manuscript. The authors should

either provide solid arguments about the fact that velocity alignment arising

from inelastic collisions is negligible in their system or explicitly say that this

point represents an important difference with respect to the system investigated

by Caprini and Löwen.

b) It is known that an effective active force can be realized with macro-

scopic objects which rectify mechanical vibrations using simple structures with asymmetric features [6]. For example, in this reference [6], a collective motion is observed for granular disks with an asymmetric base. In the submitted manuscript the authors say that the Brownian vibrators feel an active force "generated through the friction between the particle and the platform due to the unique structure of the inclined legs of the particles". I find difficult to understand how this could happen with the Brownian vibrators used in this study.

Indeed, friction alone cannot rectify mechanical vibrations but also some asymmetry of the vibrated object (or the vibrating plate) is needed. If the active forces arise from this rectification mechanism, then I cannot find a substantial difference with respect to Ref. [6] which was published more than 10 years ago.

A part from this, I find the characterization of the single particle dynamics not exhaustive. I think that more effort should put into convincing the reader that a single Brownian vibrator really performs the typical motion of an active Brownian particle as described by Eq. 2 and 3 of the main text. This could be done e.g. by comparing experimental data of single particle dynamics with some theoretical/numerical predictions for Eq. 2 and 3. From the analysis provided in the current version of the manuscript it is not easy for me to exclude that a single vibrator simply performs an ordinary (i.e. passive) underdamped Brownian motion.

2 Minor comments

2.1 Characterization of the phenomenology

a) The authors refer very often to Ref. [1], here flocking dynamics is characterized by scale-free spatial correlations of velocity polarization. I understand that demonstrating scale-free correlations in experiments is much harder than in simulations but I would suggest at least to show (in the main text or in some appendix) the behaviour of this observable (Eq. 3 of [1]) in presence of collective motion.

b) Another point which is not clear to me is the following: when performing independent experiments for one packing fraction where collective motion is observed, is the verse of the global rotation always the same? Do the authors observed changes of global motion from clockwise to counterclockwise and vice versa during a single long experimental run? I think these points should be clarified in the main text.

2.2 Miscellaneous

a) In the section "Structure", the part of the text that starts with "From theoretical work by..." and ends with "...rotation around the center of mass of the system." does not really talks about the system's structure. I would move this to a new section of the paper fully dedicated to the connection between the

experimental system under study and active Brownian particles.

b) In the section "Methods" the authors report a numerical value ($\epsilon = 0.39$) for the restitution coefficient. How did they measure it? The measure of μ is explained in Appendix A. The authors should refer to it in the main text when the numerical value of μ is provided.

c) In appendix A, one can see that the persistence time τ_p is between 0.05 and 0.25 seconds while the inertia time $\tau_I = 1.17s$. This may results confusing

since in Ref. [1] it is clearly stated that the collective motion is present only when $\tau_I \ll \tau_p$ otherwise it is suppressed. Related to this it, is not clear why τ_I

is measured in single-particle experiments while τ_p is not. At least this is what I understood from Appendix A.

d) Below Eq. A8, I found the sentence "Here, m represents the particle mass, γ is the drag coefficient" a bit confusing since in the above equation m and γ do not appear.

e) In section 2.1 of the main text: "similar to the curve of $\phi = 0.690$ in Fig.1(f).", I think the authors wanted to refer to Fig. 1(e).

References:

[1] L. Caprini and H. Löwen. Flocking without alignment interactions in attractive active brownian particles. Phys. Rev. Lett., 130:148202, Apr 2023.

[2] Gongwen Peng and Takao Ohta. Steady state properties of a driven granular medium. *Phys. Rev. E*, 58:4737–4746, Oct 1998.

[3] T. P. C. van Noije, M. H. Ernst, E. Trizac, and I. Pagonabarraga. Randomly driven granular fluids: Large-scale structure. *Phys. Rev. E*, 59:4326–4341, Apr 1999.

[4] C. Bizon, M. D. Shattuck, J. B. Swift, and Harry L. Swinney. Velocity Correlations in Driven Two-Dimensional Granular Media, pages 361–371. Springer Netherlands, Dordrecht, 2002.

[5] G. Gradenigo, A. Sarracino, D. Villamaina, and A. Puglisi. Non-equilibrium length in granular fluids: From experiment to fluctuating hydrodynamics. *Europhysics Letters*, 96(1):14004, sep 2011.

[6] Julien Deseigne, Olivier Dauchot, and Hugues Chaté. Collective motion of vibrated polar disks. *Phys. Rev. Lett.*, 105:098001, Aug 2010.

Response to referees - "Anomalous large-scale collective motion in granular Brownian vibrators"/Chen

The following significant modifications have been made to the manuscript:

(a) We have rewritten the Section 2 "Results" to more clearly and directly delineate the differences between the four phases we observed. We particularly emphasized the characteristics of particles in the flocking fluid phase, where the particles exhibit random motion on short timescales and unidirectional flocking on long timescales. This nontrivial phenomenon exhibits fundamental differences from the collective motion typically observed in systems of polar active matter, which is characterized by alignment based on instantaneous particle velocities and orientations.

(b) We have elucidated both the similarities and differences between our experiment and the simulation by Caprini et al(Ref. [32]). Instead of making arbitrary assertions that "our experiment aligns remarkably well with the theoretical prediction", we corrected the dynamics equations of active particles to fit our experiments. Then we applied the corrected equations to propose our own physical interpretation of the emergence of flocking fluid phases.

(c) We clarified the active source of Brownian vibrators and discussed its distinction from environmental Brownian noise, thus elucidating the differences between non-polar Brownian vibrators and polar active particles as well as passive particles.

(d) We have also addressed the presence of inherent asymmetry in the experimental system, such as the angle between the vibration direction and gravity, and micro-asymmetries in boundaries and particles. Importantly, we have explicitly clarified that these factors have no bearing on the emergence of collective behavior resulting from the intricate interplay between active forces and effective attractive interactions.

With these comprehensive modifications, our manuscript is now even more robust and insightful, providing a clearer understanding of the intriguing phenomena observed in our study.

Reviewer 1

In this paper, the authors are studying a system of granular particles under vertical vibrations. Different regimes of packing fraction are explored and reveal the occurrence of four distinct phases: granular fluid, collective fluid, poly-crystal, and crystal. Poly-crystal and crystal phases do not differ from equilibrium. In contrast, granular fluid and collective fluid phases are characterized by collective motion and, thus, have no equilibrium counterpart. The authors explain collective motion by noting that inelastic collisions can be roughly approximated by attractive interactions and an active force. In this way, the authors claim to have experimentally verified the numerical results of previous papers.

In my opinion, the findings of the paper are interesting: the experimental realization of collective motion numerically observed in a previous paper constitutes a novel result, while poly-crystalline and crystalline phases show findings that are rather expected. However, the sections presenting the results are technical and not easy to read. They contain several details that do not easily allow the reader to focus on the novel findings of the paper. In addition, the authors should further investigate the link between their system and the active model employed in previous studies.

A consistent major revision is needed. To be suitable for publication in Nature Communications, the paper should be rewritten to increase its clarity mainly to focus on the collective motion

observed at intermediate packing fraction. I list below several points that the authors should discuss in the new version of the paper.

Dear Reviewer 1,

Thank you very much for recognizing the novelty and interest in our work, and we greatly appreciate your suggestions on our writing details. Based on your recommendations, we have rewritten the second section of the results, removing overly intricate descriptions from the previous version. Instead, we have emphasized the qualitative differences between the newly discovered flocking fluid phase and existing experiments and theories on active particles.

Your insightful comments and valuable suggestions have played a crucial role in enhancing the clarity and completeness of our experimental description. In the following, we reply to your suggestions and comments point by point, and mark the corresponding modifications in the main text.

- in Sec. 2.1, the authors discuss the dynamics of the system for different values of the packing fraction. I have found this section unclear. Indeed, the authors start with a long description of the super-diffusive and subdiffusive behavior by varying the packing that I have, honestly, found super detail. Then, the authors continue the paragraph by quickly introducing the vorticity of the system that is used to show the presence of collective motion. While this measurement is fundamental to distinguishing between collective motion and arrested states, the authors do not focus on this point enough. By summarizing, the authors should carefully describe Fig.1(f) which contains the central result of the paper, i.e.the distinction between the four states and, in particular, the presence of the collective fluid. The authors should describe how the distinction between the four phases is obtained. I suggest a complete rewriting of Sec. 2.1.

While Omega is used to distinguish between the granular fluid phase and collective fluid, the criterion to distinguish between polycrystalline and crystalline phases is not clearly discussed.

Thank you for your invaluable feedback on our writing. Your careful review and insightful comments are sincerely appreciated.

Based on your constructive comments, we have rewritten the Sec. 2 "Results" in the revised manuscript on page 3, lines 11-13, page 4, lines 1-45, page 5, lines 1-7. After showing the snapshots of the particle configurations, we focus on the anomalous collective behavior of particles in the flocking fluid phase. This discussion leads to the introduction of the order parameter in the phase diagram - the vorticity of the displacement field, Ω , in Fig 1(f). Additionally, we added a curve that shows the variation of the order parameter ψ_6^{global} in Fig 1(f), which distinguishes between polycrystal and crystal phases with the packing fraction.

Subsequently, we introduce the diffusive properties of particles in 4 different phases through the MSD curves in Fig1(g), and removed unnecessary and overly detailed descriptions from the previous version.

Your keen observations have been immensely beneficial, significantly improving the quality of our manuscript. Your valuable input has strengthened the overall study, and we are fully committed to presenting the most accurate and rigorous research possible.

- In the literature, there are several studies based on standard granular particles, i.e. non-active. In the introduction, the authors claim that their findings are fundamentally different from those obtained in previous studies. The authors should explain the reason behind this discrepancy.

Thanks for your perceptive comments.

After rigorous screening in pre-experiments, every particle in our experiments can be regarded as identical particles. Their translational and rotational velocity distributions are Gaussian distributions with a mean of 0, which has never been achieved in previous experiments. The inelastic collision in actual experiments are too complex to accurately replicate in numerical simulations. In

the discussion section, we showed the differences between previous experiments and our Brownian vibrators one by one in the revised manuscript on page 9, lines 44-45 and page 10, lines 1-17. In Appendices A and C, we compared the pair correlation functions and the hexatic order parameters between Brownian vibrators and equilibrium hard-sphere experiments, revealing the differences in spatial structure and symmetry. In summary, the particle fluid phase and collective fluid phase observed in our experiments have not been reported in previous literature.

Once again, thank you for your thoughtful review and insightful comments, which have significantly enriched the scientific merit of our study.

- In Sec.2, the authors wrote that a single-granular particle displays passive Brownian motion characterized "by uncorrelated translation and rotation with Gaussian distributions centered around zero means" as in their previous study. How does this previous result match with the notion of persistence time that is introduced in Section 2.1?

Thank you for your incisive comment.

We have corrected an inappropriate description in the revised manuscript on page 3, line 11-13, to "In the pre-experiment, each individual particle exhibits Brownian-like behavior, characterized by uncoupled translation and rotation with Gaussian distributions centered around zero means, in the absence of particle-particle collisions". The cross-correlation between the translational and rotational velocities of our vibrating Brownian particles is close to 0, indicating the absence of coupling between translational and rotational motions. In fact, the time correlation function of the translational velocity components decays to 0 over a persistent time, which is consistent with the duration of the quasi-ballistic motion corresponding to the superdiffusion in the first part of the MSD in Fig 1(g) in the main text.

Thank you again for your careful review and thoughtful consideration. Your expertise and engagement in our work have been invaluable, and we are deeply grateful for your contributions to this research endeavor.

- The authors are proposing that inelastic collisions can be mapped onto the combination of an effective attraction and a persistent active force. Could the author further comment and intuitively justify this approximation? This is an important point that deserves to be addressed numerically. The authors should discuss this approximation more in detail, by citing references (if possible) and by showing additional studies to compare the dynamics of attractive active particles and granular passive particles with inelastic collisions. This point could be addressed numerically and it is fundamentally to claim the occurrence of the collective motion obtained in previous studies with active particles.

We genuinely appreciate the thoughtfulness and depth of your comments.

The energy loss caused by inelastic collisions between particles is a manifestation of the system's non-equilibrium nature and fundamentally differentiates our system from equilibrium hard sphere systems. The actual physical processes of inelastic collisions are complex, but it is generally accepted that the inelastic collisions between particles are equivalent to an effective attraction [Ref. 55-58]. For the first time, we propose an algorithm to quantitatively calculate this effective attraction potential by Boltzmann law.

The active force on our non-polar active particles is generated by collisions between the particle's tilted legs and the vibrating bottom plane.

And the competition between the effective mutual attraction derived from this algorithm and the active forces can reasonably explain the spontaneous emergence of flocking observed in our experiment. Specific calculation details can be found in the Appendix A.

We have not yet found numerical simulation results for purely elastic, non-polar active particles with mutual attraction potentials similar to our system. We sincerely appreciate your constructive suggestion and agree that this indeed represents a very meaningful and interesting direction for future research on inelastic collisions.

- The authors of Ref.[32] termed the collective motion observed as "flocking". Since the authors are claiming to have experimentally reproduced such a result, I suggest they use the same nomenclature. Note that, in the literature, there are other examples of experimental active granular particles where flocking is claimed even if the system is confined in a box.

Thank you for your candid suggestion. We have used "flocking" to describe the observed collective movement in the revised manuscript.

However, in the revised manuscript, we have elucidated both the similarities and differences between our experiment and the simulation by Caprini et al(Ref. [32]). Instead of making arbitrary assertions that "our experiment aligns remarkably well with the theoretical prediction", we corrected the dynamics equations of active particles to fit our experiments. Then we applied the corrected equations to propose our own physical interpretation of the emergence of flocking fluid phases.

Compared to the simulations(Ref. [32]), where collective behavior emerges when the maximum force exerted by neighboring particles surpasses the active forces of the individual particles, and the instantaneous velocity directions of particles show global polarity, our experiments did not observe the alignment effect of the instantaneous velocity directions of particles. Additionally, in the simulation the flocking phase exists under overdamped conditions, while our experiment is underdamped, making our work distinctly different from their simulation results. Due to the periodic boundary conditions in the simulation, the collective behavior appears as transitional flocking. However, in our quasi-2d system with fixed boundaries, the collective behavior manifests as global collective rotation around the center of the system.

It's a good suggestion for improving both terminology consistency and rigor. Thank you once again.

- The authors comment on the absence of phase separation in their system. Is this interpretation consistent with the model assumed to understand collective motion? In other words, is the effective Péclet number smaller than the critical Péclet number showing phase separation in systems of underdamped active particles?

Thank you for your comment. In Ref.[32], MIPS (Motility-Induced Phase Separation) occurs in repulsive-active particles, where the effective attraction force is zero, meaning the critical Péclet number is zero, smaller than the effective Péclet number. In our experiments with attractive active particles, phase separation was not observed regardless of whether the Péclet number was greater than or smaller than the critical Péclet number.

- The last paragraph of Sec. II describes the results of the intermediate scattering function at different packing fractions. This paragraph contains several technical details that should be moved in the methods. In addition, the conclusions obtained by this analysis are not clearly explained.

Thanks for your comment. Since the discussion of the intermediate scattering function are consistent with the diffusive properties shown by the MSD in Fig1(g), we decided to move this section to the Appendix F.

Minor points: - Intro: Letter – paper. - Experimental points in Fig.1 (f) are barely visible. Please, modify the figure to fix this problem.

Thank you for your careful review and suggestions. Based on your advice, we have made the corresponding modifications in the revised manuscript on page 2, line 39-40, and enhanced the symbols and curve colors in the figures to improve visibility.

Once again, we sincerely thank you for investing your time and effort in reviewing our work. We hope that the revised manuscript meets and exceeds your expectations. Should you have

any further comments or concerns, we remain open and eager to address them promptly. Your contribution has been indispensable, and we are deeply grateful for your continued engagement with our research.

Reviewer 3

The submitted manuscript reports an experimental work in which the behavior of a vertically vibrated quasi-2d granular system is explored for a wide range of packing fractions ϕ . Four different phases are observed going from low to high ϕ . The authors refer to them as granular fluid, collective fluid, poly-crystal and crystal. The paper provides an extensive characterization of the dynamics and the structure of all these phases. Among them, the main focus is given to the collective fluid where a global collective rotation is observed and put in relation with recent theoretical/numerical results on active Brownian particles. An important claim of the submitted manuscript is that this collective motion can be explained by the interplay between effective attraction and active forces. While I found the experimental methodology of this work well grounded and correctly explained, I don't find that it provides enough evidence for the observation of collective motion strictly due to the interplay between effective attraction and active forces. Thus, I can't recommend this manuscript as a publication in Nature Communications. My major concern is related to the link the authors make with the theoretical work of Caprini and Löwen [1].

Thank you for recognizing that the experimental methodology of our work well grounded and correctly explained, as well as for your detailed comments and insightful suggestions.

We have carefully considered your advice and elucidated both the similarities and differences between our experiment and the simulation by Caprini et al (Ref. [32]). In the revised manuscript, instead of making arbitrary assertions that "our experiment aligns remarkably well with the theoretical prediction", we corrected the dynamics equations of active particles to fit our experiments. Then we applied the corrected equations to propose our own physical interpretation of the emergence of flocking fluid phases.

Thanks again for your constructive suggestions. In the following, we reply to your suggestions and comments point by point, and mark the corresponding modifications in the main text. We hope that the revised manuscript meets and exceeds your expectations.

1 Major comments

1.1 Connection with active systems

The authors often refer to their system as an experimental realization of the numerical/theoretical work of Caprini and Löwen [1]. In this reference, it is shown that attractive interactions and active forces are minimal ingredients for the emergence of collective motion (i.e. flocking) characterized by long-range spatial correlations of velocity polarization. One important message of this article is that explicit alignment interactions are not needed to observe flocking. In the submitted manuscript, the authors explain that the granular system considered in their experiment, which is composed by Brownian vibrators (i.e. inelastic disks in contact with the vibrating plate through 12 randomly tilted legs), share the same key-properties of the active Brownian particles interacting via attractive potential used in [1]. An effective attraction between grains results from inelastic collisions, a persistent active force is generated through the friction between the particle and the platform due to the unique structure of the inclined legs of the Brownian vibrators. I agree that both these key-properties can potentially emerge from the mechanisms highlighted by the authors however I have two objections: a) It is generally accepted that inelastic collisions are equivalent to an effective attraction but they also introduce in the system an effective velocity alignment mechanism. Indeed, two colliding inelastic grains mainly lose energy along the direction of the unit vector that connects the particle centers. Thus, after a collision, the velocities of the two grains result more aligned than before. This mechanism can be the origin of spatial velocity correlations in granular systems [2, 3, 4, 5]. The intrinsic presence of effective velocity alignment interactions in granular dynamics is in contradiction with the claim of the authors that the emergence of flocking in their system is only due to effective attraction and active forces as in Ref. [1]. I think that this is a crucial point that should be discussed in the main text of the manuscript. The authors should either provide solid arguments about the fact that velocity alignment arising from inelastic

collisions is negligible in their system or explicitly say that this point represents an important difference with respect to the system investigated by Caprini and Löwen.

Thank you once again for your constructive suggestions and profound insights; your understanding of the related work is impressively deep. After re-examining our data based on your suggestions, it's clear that there is a qualitative difference between our experiment and the system investigated by Caprini et al(Ref.[32]).

1.The instantaneous velocity alignment effect observed in the flocking phase of Ref.[32] and Ref.[55-58] does not exist in the flocking fluid phase. As shown in Figure 1 below, we analyzed the angles $\theta_i(0 \text{ to } \pi)$ between the velocity directions of over 10,000 particle pairs 0.025s before collision and the angles $\theta_o(-\pi \text{ to } \pi)$ between their velocity directions 0.025s after collision. In both the granular fluid phase and the flocking fluid phase, we did not observe a reduction in θ_o that would indicate an instantaneous velocity alignment effect.

Figure 1: (a) $\phi = 0.270$ in granular fluid phase; (b) $\phi = 0.555$ in flocking fluid phase.

2.The spatial correlation function of the instantaneous velocities, as depicted in Figure 2(a), demonstrates that the velocity field lacks polar order in both the granular fluid phase and the flocking fluid phase. However, for $dt=10s$ and $100s$ in Figure 2(c) and (d), the spatial correlation function of the displacements shows long-range spatial correlations at $\phi = 0.555, 0.713$ and 0.832 , confirming that there is no collective motion at short time scales, and the flocking fluid phase can only be observed at time scales greater than 10s. Corresponding results have been added to Appendix H.

Here the spatial connected correlation function of the displacements is defined as:

$$C(r) = \frac{\sum_{i,j} \delta u_i \cdot \delta u_j \delta(r - r_{ij})}{\sum_{i,j} \delta(r - r_{ij})},$$

δu_i and δu_j are the displacement of the particles at positions i and j in the time interval of dt .

3.It's worth mentioning that the flocking observed in our system is different from the collective motion observed in previous experiments with active particles. Over short time scales, the collisions between Brownian vibrators are rare, and the effective mutual attraction is extremely weak, with the active forces dominating, resulting in the particles moving randomly. Only over longer time scales, as the total number of collisions between particles increases and the effective mutual attraction gradually exceeds the active forces, does the system exhibit flocking behavior. In our analysis, the effective mutual attraction potential, calculated using the pair correlation function at steady state, results from the accumulation of numerous particle collisions over a long period. Therefore, the competition between the calculated effective mutual attraction force and the active force determines the emergence of collective motion over long time scales, and is unrelated to the behavior of particles on short time scales.

Figure 2: The spatial connected correlation function of the displacements. (a) The time interval $dt=0.025s$, while it's the same as the connected correlation function of the instantaneous velocities; (b) $dt=1s$; (c) $dt=10s$; (d) $dt=100s$.

We genuinely appreciate the thoughtfulness and depth of your comments, as they have undoubtedly contributed to the refinement of our work. Your constructive feedback has motivated us to present a more solid and impactful study. We are deeply grateful for the time and effort you have invested in reviewing our manuscript. Once again, thank you for your valuable contributions to our research.

b) It is known that an effective active force can be realized with macroscopic objects which rectify mechanical vibrations using simple structures with asymmetric features [6]. For example, in this reference [6], a collective motion is observed for granular disks with an asymmetric base. In the submitted manuscript the authors say that the Brownian vibrators feel an active force "generated through the friction between the particle and the platform due to the unique structure of the inclined legs of the particles". I find difficult to understand how this could happen with the Brownian vibrators used in this study. Indeed, friction alone cannot rectify mechanical vibrations but also some asymmetry of the vibrated object (or the vibrating plate) is needed. If the active forces arise from this rectification mechanism, then I cannot find a substantial difference with respect to Ref. [6] which was published more than 10 years ago. A part from this, I find the characterization of the single particle dynamics not exhaustive. I think that more effort should be put into convincing the reader that a single Brownian vibrator really performs the typical motion of an active Brownian particle as described by Eq. 2 and 3 of the main text. This could be done e.g. by comparing experimental data of single particle dynamics with some theoretical/numerical predictions for Eq. 2 and 3. From the analysis provided in the current version of the manuscript it is not easy for me to exclude that a single vibrator simply performs an ordinary (i.e. passive) underdamped Brownian motion.

Thank you for your unique suggestions. In the revised manuscript on page 6, lines 12-21 and page 10, lines 1-17, we have provided a detailed explanation of the source of the active driving force, as well as the calculation details of the active forces in Appendix A.

The active force on our non-polar active particles is generated by collisions between the particle's tilted legs and the vibrating bottom plane. The unique design of the legs and the presence of noise cause the particle's central axis to slightly tilt away from the gravitational direction during vibration. This results in only a portion of the legs (commonly observed in experiments as a single leg) being instantaneously propelled upon making contact with the bottom surface. Such interactions generate a driving force that accelerates the particle in the horizontal direction, which is influenced by both the velocity at which the collision occurs and the angle of contact between the leg and the bottom surface. The tilted legs leads to a minimal time correlation of the contact angle, resulting in a random orientation of the driving force with a magnitude of $\gamma_0 v_0$ (Appendix A). The randomness of the driving force direction is precisely the reason why both the translational and rotational velocities of a single Brownian vibrator are Gaussian distributed with zero mean. The key distinction between this active driving force and Brownian noise lies in the orientation of one leg of the non-polar particle during each collision with the bottom plane, with the collision interval approximately 38ms and the direction switching frequency of the driving force close to our time resolution of 25ms. Therefore, the particle is driven by a fixed-magnitude, high-frequency and randomly orientation-switching active force, rather than the environmental Gaussian noise of passive particles.

In summary, the source of the active force experienced by the particles stems from the structural asymmetry during the contact between a single tilted leg and the bottom surface. However, the direction of the resultant active force is related to the current instantaneous velocity direction and the tilt angle of the particle's leg, with the switching frequency of the active force direction being very high. The active force in our experiment significantly differs from the active force along the orientation direction experienced by polar particles as described in Ref.[16] and the environmental Gaussian noise of passive particles.

2 Minor comments

2.1 Characterization of the phenomenology

a) The authors refer very often to Ref. [1], here flocking dynamics is characterized by scale-free spatial correlations of velocity polarization. I understand that demonstrating scale-free correlations in experiments is much harder than in simulations but I would suggest at least to show (in the main text or in some appendix) the behaviour of this observable (Eq. 3 of [1]) in presence of collective motion.

Thank you for your constructive suggestion. As shown in Figure 2(d) and Appendix H, for $\Delta t=100s$, the spatial correlation function of the displacements shows long-range spatial correlations at $\phi = 0.555$ in flocking fluid phase, and there is no spatial correlation at $\phi = 0.270$ in granular fluid phase. Figure 2(d) is similar to the Fig. 3(a) in Ref.[32], which is the spatial correlation function of the instantaneous velocities and shows the difference between the flocking phase and the disordered phase. However, there is no long-range order shown in the spatial correlation function of the instantaneous velocities in our Figure 2(a).

b) Another point which is not clear to me is the following: when performing independent experiments for one packing fraction where collective motion is observed, is the verse of the global rotation always the same? Do the authors observed changes of global motion from clockwise to counterclockwise and vice versa during a single long experimental run? I think these points should be clarified in the main text.

Thank you for your careful consideration. We have conducted a thorough reevaluation of our experimental procedures, taking into account possible sources of asymmetry in the setup that could influence the observed rotation direction. As dedicated researchers, we strive for the highest level of control and accuracy in our experiments to ensure the validity and reliability of our findings.

It is true that inherent asymmetries do exist in the experiment, such as the micro angle between the vibration direction along the z-axis and the force of gravity, as well as micro asymmetries in the boundaries and particles. However, we would like to emphasize that the emergence of collective behavior is primarily a result of the competition between active forces and effective attractive interactions, rather than being solely driven by the system's inherent asymmetry. Otherwise, collective behavior would be expected to occur at any packing fraction, especially for $\phi \leq 0.270$, where particle collisions are less frequent, potentially magnifying the effects of setup asymmetry.

In our experiments, we have made careful observations and noted that for certain conditions, particularly when the packing fraction is $\phi \leq 0.270$, the particles exhibit random and disordered motion within a time scale of 1 hour. However, as the volume fraction increases and reaches a critical point at approximately 0.317, collective behavior emerges, characterized by clockwise rotation around the symmetry center of the system. This rotational behavior can be attributed to the interplay between active forces and effective attractive interactions, leading to the observed collective motion.

Moreover, it is crucial to mention that the continuous experimental measurements were conducted under identical conditions across different trials, ensuring comparability among groups with varying packing fractions. As a result, the presence of clockwise rotation in the collective fluid phase is consistently observed. However, during the preliminary experiments and the setup calibration period, we have also observed instances of counterclockwise rotation.

Overall, our experimental data analysis shows that the competition between active forces and effective attractive interactions plays a crucial role in the emergence of collective behavior, outweighing the influence of inherent system asymmetry. We thank the reviewer for their valuable comments, and in response, we have provided additional clarifications in the revised manuscript on page 4, lines 17-20.

2.2 Miscellaneous

a) In the section "Structure", the part of the text that starts with "From theoretical work by..." and ends with "...rotation around the center of mass of the system." does not really talks about

the system's structure. I would move this to a new section of the paper fully dedicated to the connection between the experimental system under study and active Brownian particles.

Thank you for your detailed review and constructive feedback. After carefully considering the logic and coherence of the narrative, we decided to remove the title of subsections on dynamics and structure. In Fig. 1(f) in the main text, we introduced the hexagonal order parameter characterized by structure. Following that, in Fig. 2(b), we discussed the first anomalous peak of the pair correlation function $g(r)$ for dilute cases, which naturally leads to the discussion related to Fig. 3 on the equivalent attraction force and the dynamics equations for active particles. Finally, based on the structural information obtained earlier, we discuss the anomalous fluctuations in particle number density shown in Figure 4.

Thank you again for your suggestions, which have greatly enhanced the logic and coherence of our paper.

(b) In the section "Methods" the authors report a numerical value ($\mu = 0.39$) for the restitution coefficient. How did they measure it? The measure of μ is explained in Appendix A. The authors should refer to it in the main text when the numerical value of μ is provided.

Thanks for your helpful suggestions. We added the reference in the revised manuscript on page 9, lines 25.

(c) In appendix A, one can see that the persistence time τ_p is between 0.05 and 0.25 seconds while the inertia time $\tau_I = 1.17$ s. This may result in confusing since in Ref. [1] it is clearly stated that the collective motion is present only when $\tau_I \ll \tau_p$ otherwise it is suppressed. Related to this it is not clear why τ_I is measured in single-particle experiments while τ_p is not. At least this is what I understood from Appendix A.

Thank you for your constructive comment. Here $\tau_I \ll \tau_p$ represents a significant difference between the long-term scale flocking observed in our experiments and the flocking based on instantaneous velocity alignment effects seen in the simulations of ref.[32]. In our experiments, the particle motion is underdamped, and indeed, collective motion is not observed on short time scales, which is consistent with the assertions in ref.[32]. However, the existence of flocking on long time scales was not discussed in ref.[32]. In fact, due to the fact that the experimental setup where individual particles are randomly driven is seriously influenced by the gravitational drift, previous works have not thoroughly discussed the collective behavior of particles over long timescales. Our experiments are the first to achieve data collection for randomly driven particles on a timescale ranging from 0.025 seconds to hours.

In this paper, the introduction of the single particle's (τ_I) is to calculate the active driving force ($\gamma_0 v_0$). Considering that the source of the active driving force is the collision between the particle and the bottom surface, and this collision frequency is much higher than that of particle-particle collisions (Appendix G), the active driving force is independent of the packing fraction. Therefore, calculating the simplest τ_I of a single particle is sufficient to determine the magnitude of the active driving force. However, the duration τ_p decreases with an increase in volume fraction, and we need the τ_p at different packing fractions to calculate the Péclet number (Pe). Therefore, we have provided the τ_p for each packing fraction in Appendix A.

Thanks again for your thoughtful comments.

(d) Below Eq. A8, I found the sentence "Here, m represents the particle mass, ζ is the drag coefficient" a bit confusing since in the above equation m and ζ do not appear.

Thank you for your careful review. We have removed the superfluous descriptions in Appendix A.

(e) In section 2.1 of the main text: "similar to the curve of $\beta = 0.690$ in Fig.1(f).", I think the authors wanted to refer to Fig. 1(e).

Thanks again for your careful review. We have corrected this typographical error in the revised main text on page 4, line 29.

With your guidance, we are confident that the revised manuscript now offers a more robust and compelling contribution to the field. We eagerly look forward to your further evaluation, and we hope that the updated version will be deemed suitable for publication on Nature Communications. Your thoughtful feedback has been indispensable in refining our research, and we sincerely appreciate your valuable contribution to our work.

REVIEWER COMMENTS

Reviewer #1 (Remarks to the Author):

In the new version of the paper, the authors have mainly addressed my doubts and concerns.

However, unluckily, I have a last concern to raise before I can accept the paper for publication.

In my previous reply, I wrote:

"In my opinion, the findings of the paper are interesting: the experimental realization of collective motion numerically observed in a previous paper constitutes a novel result, while polycrystalline and crystalline phases show findings that are rather expected."

Indeed, it is well known that fluid and crystalline phases can be observed in granular systems in the absence of activity and, in addition, these phases are well studied in simulations. The novelty of the paper consists of the observation of the flocking fluid phase arising from the activity. Even if flocking is not new in active matter, this collective motion has been experimentally observed for elongated particles (which tend to align for entropic reasons) or with particles subject to alignment interactions, in previous papers. Here, particles are not elongated and there is no trace of alignment interactions. Additionally, it is known that flocking cannot be observed in passive granular particles. These points should be stressed in the paper.

In addition, I have noted a change in the presentation of the results on which I do not agree. After noting that flocking was recently predicted for spherical active particles without alignment interaction (Ref. [32]), in the previous version of the paper the authors claimed to have experimentally confirmed these results. Previously, the authors wrote a sentence that is currently removed:

"Remarkably, our experiments demonstrate the occurrence of large-scale collective particle motion within the range of $0.317 \leq \phi < 0.713$, which aligns remarkably well with the theoretical prediction made by Caprini et al.[32]."

Now, the authors now uniquely focus on the quantitative difference with Ref.[32]. In spite of these quantitative differences, I believe that the present paper has to be presented as an experimental verification of Ref.[32], since, in both cases, active particles show collective motion even in the absence of alignment interactions. The authors should comment on this point, as in the previous version.

The revised version of the manuscript, together with the author's reply, goes in the right direction in resolving my doubts. I find the answer to major comment 1.1a satisfactory, but I can't say the same for major comment 1.1b. Thus, I can't still recommend the revised version of the manuscript for publication in Nature Communications. Here are the points that should be addressed before reconsidering the paper for publication.

1 Major comments

1) It is hard for me to understand which model is the correct one to describe the dynamics of a *single* isolated Brownian vibrator. Indeed, the way the authors refer to Eq. 2 and 3 is not clear. On one hand, it seems they use them to describe the motion of an isolated particle, on the other hand, they plot the Peclet number $= v_0/(D_r * D)$ as a function of the packing fraction making me think that they are considering the model parameters as effective parameters which include the effect of particle interactions. If v_0 and D_r are parameters of the model for the single particle dynamics and D is the particle radius how does a quantity like $v_0/(D_r * D)$ can vary with the packing fraction?

2) With this sentence in the reply

"Therefore, the particle is driven by a fixed-magnitude, high-frequency and randomly orientation-switching active force, rather than the environmental Gaussian noise of passive particles. In summary, the source of the active force experienced by the particles stems from the structural asymmetry during the contact between a single tilted leg and the bottom surface. However, the direction of the resultant active force is related to the current instantaneous velocity direction and the tilt angle of the particle's leg, with the switching frequency of the active force direction being very high. The active force in our experiment significantly differs from the active force along the orientation direction experienced by polar particles as described in Ref.[16] and the environmental Gaussian noise of passive particles."

the authors somehow define a model for the isolated Brownian vibrator by words. It would be extremely clarifying if they could express by equation the model they have in mind when they talk about *"a fixed-magnitude, high-frequency and randomly orientation-switching active force"* and explain how it differs from the one for *"the active force along the orientation direction"*. Eq. 2 and 3 provided in the text describe this second condition: the single particle is described by the velocity v and the orientation θ . The active force $\gamma_0 v_0$ acts along the orientation. So, if the authors are using these two equations to describe the motion of isolated particles, they are in contradiction with what is stated in the reply.

Summarizing, which is the stochastic model describing the motion of an isolated Brownian vibrator? How does it differ from the standard underdamped Brownian motion? How does it differ from the active Brownian particle model

Eq. 2 and 3? Are data of standard observables like MSD and displacement distribution obtained from isolated particles in agreement with the proposed model?

2 Minor comments

I still find the overall structure of the paper a bit cumbersome. To make the paper more readable and more focused on the collective motion I would move as much as possible the analysis of the structure of non-flocking phases to an appendix. In my opinion, the only relevant structural analysis important for the main text is the one that shows the connection between the deviation of the $g(r)$ from equilibrium and the effective attraction.

Response to referees - "Anomalous large-scale collective motion in granular Brownian vibrators"/Chen

Reviewer 1

In the new version of the paper, the authors have mainly addressed my doubts and concerns. However, unluckily, I have a last concern to raise before I can accept the paper for publication.

Dear Reviewer 1,

Thank you for recognizing our revised manuscript and for your constructive comments. In the following text, we will revise each point according to your suggestions.

In my previous reply, I wrote: "In my opinion, the findings of the paper are interesting: the experimental realization of collective motion numerically observed in a previous paper constitutes a novel result, while polycrystalline and crystalline phases show findings that are rather expected." Indeed, it is well known that fluid and crystalline phases can be observed in granular systems in the absence of activity and, in addition, these phases are well studied in simulations. The novelty of the paper consists of the observation of the flocking fluid phase arising from the activity. Even if flocking is not new in active matter, this collective motion has been experimentally observed for elongated particles (which tend to align for entropic reasons) or with particles subject to alignment interactions, in previous papers. Here, particles are not elongated and there is no trace of alignment interactions. Additionally, it is known that flocking cannot be observed in passive granular particles. These points should be stressed in the paper.

Thank you for highlighting the significance and novelty of our work. According to your suggestions, we have revised the lines 27-29 on page 2, lines 3-10 on page 3, lines 33-34 on page 6 and lines 1-15 on page 7 of the manuscript, emphasizing the fact that flocking cannot be observed in passive granular particles and the differences between the global flocking we observed and the previously reported elongated particle flocking with alignment effects.

In addition, I have noted a change in the presentation of the results on which I do not agree. After noting that flocking was recently predicted for spherical active particles without alignment interaction (Ref. [32]), in the previous version of the paper the authors claimed to have experimentally confirmed these results. Previously, the authors wrote a sentence that is currently removed: "Remarkably, our experiments demonstrate the occurrence of large-scale collective particle motion within the range of $0.317 \leq \phi < 0.713$, which aligns remarkably well with the theoretical prediction made by Caprini et al.[32]." Now, the authors now uniquely focus on the quantitative difference with Ref.[32]. In spite of these quantitative differences, I believe that the present paper has to be presented as an experimental verification of Ref.[32], since, in both cases, active particles show collective motion even in the absence of alignment interactions. The authors should comment on this point, as in the previous version.

Thank you for your constructive comments. Based on your suggestions, we have revised the lines 3-10 on page 3, lines 33-34 on page 6 and lines 1-15 on page 7 of the manuscript to emphasize

that our experimental results confirm the existence of flocking behavior in non-polar active matter in the absence of alignment interactions, as predicted by Caprini et al. in reference [32].

Once again, we sincerely thank you for dedicating your time and effort to reviewing our work. We hope that the revised manuscript meets and exceeds your expectations. If you have any further comments or concerns, we are more than willing to address them promptly. Your contribution has been invaluable, and we greatly appreciate your ongoing involvement with our research.

Reviewer 3

The revised version of the manuscript, together with the author's reply, goes in the right direction in resolving my doubts. I find the answer to major comment 1.1a satisfactory, but I can't say the same for major comment 1.1b. Thus, I can't still recommend the revised version of the manuscript for publication in Nature Communications. Here are the points that should be addressed before reconsidering the paper for publication.

Dear Reviewer 3,

Thank you for recognizing that our revised manuscript and previous responses are going in the right direction, and for your satisfaction with our response to comment 1.1a. We apologize that our response to comment 1.1b did not fully address your concerns. In the following, we will carefully address each of your comments and make the corresponding changes in the manuscript.

In the following, we reply to your suggestions and comments point by point, and mark the corresponding modifications in the main text. We hope that the revised manuscript meets and exceeds your expectations.

1 Major comments

1) It is hard for me to understand which model is the correct one to describe the dynamics of a single isolated Brownian vibrator. Indeed, the way the authors refer to Eq. 2 and 3 is not clear. On one hand, it seems they use them to describe the motion of an isolated particle, on the other hand, they plot the Peclet number $= v_0/(D_r * D)$ as a function of the packing fraction making me think that they are considering the model parameters as effective parameters which include the effect of particle interactions. If v_0 and D_r are parameters of the model for the single particle dynamics and D is the particle radius how does a quantity like $v_0/(D_r * D)$ can vary with the packing fraction?

Thanks for your detailed comment. The equations [2] and [3] in this manuscript are referenced from the simplified dynamical equations for non-polar active particles in reference [32]. These equations describe the motion of any particle i in a multi-particle system, where D_r is the diffusion coefficient of the particles at a given packing fraction. As the packing fraction increases, the collision frequency between particles increases, resulting in a decrease in the persistent distance v_0/D_r , and thus the Péclet number Pe gradually decreases (as shown in Figure 3 of the main text). The term \mathbf{F}_i represents the net force on the particle i from all other particles in the system. In our model, it is averaged to the effective attraction corresponding to the energy loss due to inelastic collisions between particles. As the packing fraction increases, the collision frequency between particles increases, which increases the effective attraction \mathbf{F}_i . We have revised the lines 7-9 and 24-28 on page 6 of the manuscript to make the corresponding description clearer.

2) With this sentence in the reply

"Therefore, the particle is driven by a fixed-magnitude, high-frequency and randomly orientation-switching active force, rather than the environmental Gaussian noise of passive particles. In summary, the source of the active force experienced by the particles stems from the structural asymmetry during the contact between a single tilted leg and the bottom surface. However, the direction of the resultant active force is related to the current instantaneous velocity direction and the tilt angle of the particle's leg, with the switching frequency of the active force direction being very high. The active force in our experiment significantly differs from the active force along the orientation direction experienced by polar particles as described in Ref.[16] and the environmental Gaussian noise of passive particles." The authors somehow define a model for the isolated Brownian vibrator by words. It would be extremely clarifying if they could express by equation the model they have in mind when they talk about "a fixed-magnitude, high-frequency and randomly orientation-switching active force" and explain how it differs from the one for "the active force

along the orientation direction". Eq. 2 and 3 provided in the text describe this second condition: the single particle is described by the velocity \mathbf{v} and the orientation \cdot . The active force $\gamma_0 v_0$ acts along the orientation. So, if the authors are using these two equations to describe the motion of isolated particles, they are in contradiction with what is stated in the reply.

Thank you for your comment. We have revised the lines 13-17 on page 6 of the manuscript to make the corresponding description clearer.

Equation [2] in the main text describes the dynamics of the particles in our experiment. Since the particles are non-polar, the direction \mathbf{n}_i of the active force is **not** the orientation of the particle. This is similar to the simulations in reference [32].

Specifically, the direction \mathbf{n}_i of the active force is determined by an inclined leg which is instantaneously propelled after contacting the bottom surface. Due to the randomness of the contact angle between the inclined legs and the bottom surface, the direction \mathbf{n}_i is **independent** of the velocity direction $\frac{\mathbf{v}_i}{|\mathbf{v}_i|}$. The direction \mathbf{n}_i of the active force changes at the same frequency with the collisions between the particles and the bottom surface.

Summarizing, which is the stochastic model describing the motion of an isolated Brownian vibrator? How does it differ from the standard underdamped Brownian motion? How does it differ from the active Brownian particle model Eq. 2 and 3?

Thanks for your constructive questions. Setting $\mathbf{F}_i = 0$ in equation [2] and [3] of the main text yields the dynamic equations of a single isolated Brownian vibrator:

$$m\dot{\mathbf{v}}_i = -\gamma\mathbf{v}_i + \gamma_0 v_0 \mathbf{n}_i + \sqrt{2\Lambda\gamma}\eta_i, \quad (1)$$

$$\dot{\theta}_i = \sqrt{2D_r}\Theta_i. \quad (2)$$

Here the difference between the dynamic equations of a single isolated Brownian vibrator and the standard passive Brownian motion lies in the active force term $\gamma_0 v_0 \mathbf{n}_i$. The distinction between the dynamic equations of a single isolated Brownian vibrator and the active Brownian particle mode equations [2] and [3] lies in the inter-particle interaction force term \mathbf{F}_i .

Are data of standard observables like MSD and displacement distribution obtained from isolated particles in agreement with the proposed model?

Thanks for your constructive questions again. Yes, they are. Due to the fact that the direction \mathbf{n}_i of the active force changes at a high frequency with the collisions between the particles and the bottom surface, a single isolated Brownian vibrator should undergo Brownian motion in the plane. The slope of the MSD curve obtained from isolated Brownian vibrators(Fig.1(g) in the main text, blue triangle) is equal to 1, and the displacement distribution(Fig.1(b) in Ref.[52]) obtained from isolated Brownian vibrators are Gaussian distribution, which are characteristics of classical Brownian motion.

2 Minor comments I still find the overall structure of the paper a bit cumbersome. To make the paper more readable and more focused on the collective motion I would move as much as possible the analysis of the structure of non-flocking phases to an appendix. In my opinion, the only relevant structural analysis important for the main text is the one that shows the connection between the deviation of the $g(r)$ from equilibrium and the effective attraction.

Thank you again for your suggestions to improve the readability of our paper. We have incorporated your suggestions(See lines 13-20 on page 5) and moved the structural information of non-flocking motion to the appendix I.

Thanks again for your detailed comments. Your thorough exploration has enhanced the rigor of our work. We eagerly look forward to your further evaluation, and we hope that the updated version will be deemed suitable for publication on Nature Communications. We sincerely appreciate your valuable contribution to our work.

REVIEWERS' COMMENTS

Reviewer #1 (Remarks to the Author):

In the new version of the manuscript, the authors have addressed my criticisms and improved the paper. Therefore, I consider the paper suitable for publication in Nature Communications in its current form.

Reviewer #3 (Remarks to the Author):

The authors addressed all my concerns about the manuscript and I now suggest it for publication in Nature Communication.

I only have one last comment:

Thanks to the authors' last reply, I understood the source of confusion around the model Eq. 2 and 3 used in the manuscript: it can be used both to describe polar active particles and non-polar ones. The difference lies in the relationship between the orientation of the active force n_i and the particle orientation. In the polar case, n_i coincides with the particle orientation (e.g. in ref 16 the vector is identified by the asymmetric shape of the disk's legs). In contrast, in the non-polar case treated in the manuscript, n_i is related to the specific impact kinematics between the randomly oriented legs of the disk and the bottom plate.

Thus, the change of orientation of the active force in a polar disk requires an actual rotation of the disk while in the non-polar case, a single particle can experience an active force with different orientations without rotating if the impact with the plate occurs with differently oriented legs.

I suggest the authors clarify this point to avoid confusion in the main text stressing the difference between cases where, as in ref. 16, n_i coincides with the particle orientation and the case under study in the submitted paper.

I also noted that the authors never define θ_i . It is related to the components of the vector $\mathbf{n}_i = (\cos \theta_i, \sin \theta_i)$. This should appear in the main text as normally done when defining abp models see, for example, ref. [32] above Eqs. 1

Response to referees - "Anomalous large-scale collective motion in granular Brownian vibrators"/Chen

Reviewer 1

In the new version of the manuscript, the authors have addressed my criticisms and improved the paper. Therefore, I consider the paper suitable for publication in Nature Communications in its current form.

Dear Reviewer 1,

Thank you for recognizing that our manuscript is suitable for publication in Nature Communications. Your constructive comments and in-depth insights have greatly enhanced the scientific quality, rigor, and readability of our work.

Reviewer 3

The authors addressed all my concerns about the manuscript and I now suggest it for publication in Nature Communication.

Dear Reviewer 3,

Thank you for considering our manuscript suitable for publication in Nature Communications. Your constructive feedback and thorough review have significantly improved the scientific quality, rigor, and readability of our work.

I only have one last comment: Thanks to the authors' last reply, I understood the source of confusion around the model Eq. 2 and 3 used in the manuscript: it can be used both to describe polar active particles and non-polar ones. The difference lies in the relationship between the orientation of the active force n_i and the particle orientation. In the polar case, n_i coincides with the particle orientation (e.g. in ref 16 the vector is identified by the asymmetric shape of the disk's legs). In contrast, in the non-polar case treated in the manuscript, n_i is related to the specific impact kinematics between the randomly oriented legs of the disk and the bottom plate. Thus, the change of orientation of the active force in a polar disk requires an actual rotation of the disk while in the non-polar case, a single particle can experience an active force with different orientations without rotating if the impact with the plate occurs with differently oriented legs. I suggest the authors clarify this point to avoid confusion in the main text stressing the difference between cases where, as in ref. 16, n_i coincides with the particle orientation and the case under study in the submitted paper.

Thanks for your constructive comments. We have revised the relevant content on page 6, lines 13-18 and 22-24 (in the third paragraph below Equation 3) of the manuscript based on your suggestions, emphasizing that the particles in our experiment are fundamentally different from the polar particles described in reference 16.

I also noted that the authors never define θ_i . It is related to the components of the vector $n_i = (\cos \theta_i, \sin \theta_i)$. This should appear in the main text as normally done when defining abp models see, for example, ref. [32] above Eqs. 1

Thanks for your detailed suggestions. We have revised the relevant content on page 6, lines 19-22 (in the third paragraph below Equation 3) of the manuscript based on your suggestions.